# RETRACTED: Regulation of Long Non-Coding RNA-Dreh Involved in Proliferation and Migration of Hepatic Progenitor Cells during Liver Regeneration in Rats

**DOI:** 10.3390/ijms20102549

**Published:** 2019-05-24

**Authors:** Zhiyan Ruan, Manxiang Lai, Ling Shang, Xiangliang Deng, Xinguo Su

**Affiliations:** 1School of Pharmacy, Guangdong Food & Drug Vocational College, Guangzhou 510520, China; ruanzy@gdyzy.edu.cn (Z.R.); laimx@gdyzy.edu.cn (M.L.); shangl@gdyzy.edu.cn (L.S.); 2School of Chinese Medicine, Guangdong Pharmaceutical University, Guangzhou 510006, China; dxl@gdpu.edu.cn

**Keywords:** long non-coding RNA-Dreh, liver regeneration, hepatic progenitor cell, liver injury, vimentin, albumin, alpha fetoprotein

## Abstract

Liver regeneration plays a significant role in protecting liver function after liver injury or chronic liver disease. Long non-coding RNAs (lncRNAs) are considered to be involved in the proliferation of hepatocytes and liver regeneration. Therefore, this study aimed to explore the effects of LncRNA-Dreh on the regulation of hepatic progenitor cells (HPCs) during liver regeneration in rats. Initially, the rat model of liver injury was established to investigate the effect of LncRNA-Dreh down-regulation on liver tissues of rats with liver injury. Subsequently, HPCs line WB-F344 cells were transfected with interference plasmid of LncRNA-Dreh and the expression of LncRNA-Dreh and Vimentin was detected. The proliferation and migration ability of WB-F344 cells, as well as the content of albumin (ALB) and alpha fetoprotein (AFP) in cell differentiation were then determined. Disorderly arranged structure of liver tissue, a large number of HPCs set portal area as center extended to hepatic lobule and ductular reaction were observed in liver tissues of rats with liver injury. The expression of LncRNA-Dreh decreased while Vimentin increased in liver tissues of rats with liver injury. Moreover, the proliferation and migration ability, expression of Vimentin and AFP in WB-F344 cells were increased after silencing of LncRNA-Dreh and ALB was decreased. Collectively, our findings demonstrate that inhibition of LncRNA-Dreh can enhance the proliferation and migration abilities of HPCs in liver regeneration but cause abnormal differentiation of HPCs.

## 1. Introduction

As the largest solid organ in the body, the liver has the functions of bile secretion, synthesis of important proteins and detoxification and when massive liver parenchymal cells are lost during liver injury or liver resection, the liver would use its powerful ability to regenerate it [1]. Liver regeneration might be defined as compensatory hyperplasia, it means that the remaining liver tissue extend to meet the needs of body metabolism [2]. Hepatic progenitor cells (HPCs) are considered to be a facultative reservoir for liver epithelial cells in healthy adult liver. HPCs are a small population of undifferentiated cells but when hepatocyte proliferation is inhibited, HPCs are activated to participate in liver regeneration [3]. During liver regeneration, HPCs are dismissed as the second line of defense. The liver has great potential to regenerate by replication of remaining healthy hepatocytes but when the replacement of inactive hepatic mass by remaining hepatocytes is partially or completely eliminated, HPCs would induce proliferation and differentiation into hepatocytes [4]. Nowadays, accumulating evidence indicate that long non-coding RNAs (LncRNAs) is involved in a variety of physiological and pathophysiological processes such as rat liver regeneration and could regulate gene expression at the transcriptional and post transcriptional levels [5]. 

LncRNAs are generally defined as non-coding RNAs of length above 200 nucleotides, which play an important role in a variety of cellular processes via interaction with main component proteins in the gene regulatory system and the alteration of their cell- or tissue-specific expression and their primary or secondary structure is regarded to promote cell proliferation, metastasis and invasion [6,7]. A study has shown that LncRNA-Dreh, down-regulated by hepatitis B virus X protein (HBx) can inhibit hepatocellular carcinoma growth and metastasis in vitro and in vivo as well as act as a tumor suppressor in the development for hepatitis B virus-hepatocellular carcinoma [8]. LncRNA-Dreh could also combine with the intermediate filament protein Vimentin and suppress its expression and thereby change the normal cytoskeleton structure to prevent tumor metastasis [9]. In addition, overexpression of Vimentin in cancer relates well to expedited tumor growth, invasion and poor prognosis, it also serves as an attractive potential target for future cancer therapy [10]. However, so far, the mechanism of LncRNA-Dreh in liver regeneration has not been elucidated. In this study, we aim to investigate the role of LncRNA-Dreh on regulation of HPCs during liver regeneration in rats. 

## 2. Results 

### 2.1. The Morphological Changes of Rat HPCs

HPCs are activated to participate in liver regeneration when hepatocyte proliferation is damaged. Initially, we compared the morphology of liver tissue in rats with liver injury, which is the basis of HPCS inducing proliferation and differentiation of hepatocytes. The results of hematoxylin-eosin (HE) staining showed (Figure 1) the liver cells in the normal group were orderly arranged in a polygonal shape with the central vein arranged radially. The morphology of HPCs in the negative control (NC) group was similar to the normal group but liver tissue structure was disorderly arranged in the liver injury group and cells varied in size with irregular arrangement, meanwhile, ballooning degeneration was observed and a large number of HPCs (oval cells) set portal area as center extended to hepatic lobule and ductular reaction was observed.

### 2.2. LncRNA-Dreh is Down-Regulated in Liver Tissues of Rats with Liver Injury

Subsequently, the expression of LncRNA-Dreh in each group was determined by RT-qPCR. The results of RT-qPCR showed (Figure 2) LncRNA-Dreh expression of liver tissue of rats had no significant difference between the normal and NC groups (*p* > 0.05) but LncRNA-Dreh expression of the liver injury group was significantly lower than that in the normal and NC groups (*p* < 0.05). These findings indicated that the expression of LncRNA-Dreh decreased in rats during liver injury.

### 2.3. Vimentin is Up-Regulated in Liver Tissues of Rats with Liver Injury

Then, the expression of Vimentin in each group was evaluated by RT-qPCR and Western blot analysis (Figure 3A–C). Compared with the NC group, the mRNA and protein expression of Vimentin in liver tissue of the normal group showed no significant difference (*p* > 0.05) but the expression in the liver injury group was significantly higher than the NC and normal groups (*p* < 0.05). These came to a demonstration that elevation of Vimentin was exhibited in rats during liver injury. 

### 2.4. Silencing of LncRNA-Dreh Promotes Proliferation of HPCs

In order to figure out the effect of LncRNA-Dreh on HPCs line WB-F344 cells proliferation after silencing, the CCK-8 assay was performed. The results of CCK-8 assay indicated that cells in each group after transfection proliferated over time (Figure 4), the proliferation rate of the blank group and the liver tissues of rats with liver injury exhibit decreased LncRNA-Dreh group at each time point had no significant difference (both *p* > 0.05). After transfection for 24 h, compared with the blank group, the proliferation rate of the si-Dreh group had no significant difference (*p* > 0.05), however, the proliferation rate of the si-Dreh group significantly increased after transfection for 48 h (*p* < 0.05). It was suggested that, suppressed LncRNA-Dreh expression could promote the proliferation of HPCs line WB-F344 cells. 

### 2.5. Silencing of LncRNA-Dreh Promotes Migration of HPCs

Aiming to figure out the effect of LncRNA-Dreh on HPCs line WB-F344 cells proliferation after silencing, the Transwell assay was conducted. The results of Transwell assay were shown in Figure 5. There was no significant difference between the number of migrated cells of blank group and the control group (*p* > 0.05). The number of migrated cells of the si-Dreh group were significantly higher than that in the blank group and the control groups (*p* < 0.05). These results implied that the inhibition of LncRNA-Dreh could enhance the migration of HPCs line WB-F344 cell.

### 2.6. Silencing of LncRNA-Dreh Promotes the Vimentin Expression of HPCs

The expression of Vimentin in HPCs line WB-F344 cells was assessed by RT-qPCR and Western blot analysis after silencing of LncRNA-Dreh. The results of RT-qPCR and Western blot analysis is shown in Figure 6A–C. It shows that the mRNA and protein expression of Vimentin in the control and blank groups did not differ significantly (*p* > 0.05) but the mRNA and protein expression of Vimentin in the si-Dreh group significantly increased compared with these two groups (*p* < 0.05). From these results, it was clear that, silencing of LncRNA-Dreh had a promoting effect on up-regulating the expression of Vimentin in HPCs line WB-F344 cells. 

### 2.7. Silencing of LncRNA-Dreh Decreases Synthesis of ALB and Enhances Synthesis of AFP in HPCs during Differentiation

The effect of silencing of LncRNA-Dreh on synthesis of ALB and AFP in HPCs was determined by using ALB kit and AFP kit. With the differentiation of WB-F344 cells, cells of the blank group began to synthesize and secrete ALB on the 6th day and reached the highest level on the 12th day. It was suggested that, rat HPCs could differentiate into hepatocytes which have the ability to secrete ALB after one week of induced culture. Compared with the blank group, the control group had no significant difference in secretion expression of ALB (*p* > 0.05) but obviously decreased level was found in the si-Dreh group (*p* < 0.05), (Figure 7A). With the differentiation of WB-F344 cells, AFP expression in the blank group gradually decreased. There was no significant difference between the blank and control groups (*p* > 0.05) but AFP was significantly increased in the si-Dreh group when compared with the blank group (*p* < 0.05) (Figure 7B). Our results pointed out that inhibition of LncRNA-Dreh could promote the ability of HPCs from differentiating into hepatocytes according to the exhibitions of decreased synthesis of ALB and increased synthesis of AFP during the differentiation of WB-F344 cells.

## 3. Discussion

If placed in a toxic environment, the liver is prone to be affected by a variety of toxins, infections, tumors and disorders, fortunately, after different types of injury, the liver has a remarkable ability to regenerate [11]. Liver regeneration is a series of physiopathological phenomena which bring about quantitative recovery from the liver mass loss to compensate reduced hepatic volume and impaired function [12]. Liver injury prevent the proliferation of mature hepatocytes and lead to the activation of HPCs, which is involved in the liver tissue repair and HPCs as bi-potential cells can be conducive to form hepatocytes and cholangiocytes when mature hepatocytes regeneration is damaged [13]. In our study, we investigated that the effects of LncRNA-Dreh on regulation of HPCs during liver regeneration in rats and found that inhibition of LncRNA-Dreh can enhance the proliferation and migration of HPCs in liver regeneration but cause abnormal differentiation of HPCs.

From this study, we could find that the liver cells were disorderly arranged in rats with liver injury, with a large number of HPCs set portal area as center extended to hepatic lobule and the ductular reaction (DR) appeared. A relevant research found that in chronic or severe liver injury, DR appeared firstly and from the portal area expanded to the hepatic parenchyma even in severe liver injury [14]. Other studies showed that in chronic injury or impaired adult liver regeneration, the bi-potent HPCs was activated and can regenerate both hepatocytes and cholangiocytes [15]. These studies presented that HPCs proliferate and differentiate into hepatocytes and cholangiocytes in the liver injury or liver resection to participate in the liver regeneration [16]. Then the comparison of LncRNA-Dreh expression of liver tissue of rats among three groups reflected LncRNA-Dreh expression of rats with liver injury which was significantly lower than the normal and NC groups. LncRNAs played key roles in a variety of biological and cellular processes and it is involved in the interaction with chromatin remodeling complexes, which are further reported by the fact that dysregulation of LncRNAs are related to many human cancers [17,18]. There was a relevant research which revealed that LncRNA-HOTAIR expression was up-regulated in primary and metastatic breast cancer, it indicated that LncRNA has been demonstrated to be involved in cancer metastasis [19]. LncRNA-Dreh as a tumor suppressor can inhibit hepatocellular carcinoma metastasis and it is also reported that inhibition of LncRNA-Dreh expression can promote the proliferation and migration of liver cancer cells [9].

Furthermore, our study presented that the mRNA and protein expression of Vimentin was significantly higher in the liver injury group than that in other groups, which suggested that the overexpression of Vimentin could cause the proliferation of HPCs. Vimentin, a type III intermediate filament (IF) protein, is majorly expressed in mesenchymal cells and it plays an important role in physiological and cellular process, including wound healing, metastasis, cell migration and focal adhesion assembly [20]. There was a study which showed that Vimentin regulated epithelial-to-mesenchymal transition (EMT) to induce migration by up-regulating the expression of receptor tyrosine kinase Axl [21]. Another study presented that, down-regulated LncRNA-BANCR conduce the up regulation of E-cadherin and down regulation of Vimentin protein expression and down regulated Vimentin expression is correlated with suppressed HCC migration [22]. In addition, Huang et al. has reported that LncRNA-Dreh unites with Vimentin protein and inhibited its expression to prevent tumor metastasis [9].

## 4. Materials and Methods 

### 4.1. Ethical Statements

The experiment was approved by Laboratory Animal Ethics Committee of Guangdong Pharmaceutical University (approval code: GDPULAC2018125; approval date: 22 October 2018). All painful procedures were performed under anesthesia and all efforts were made to minimize suffering.

### 4.2. Model Establishment

A total of 30 healthy male stable disease (SD) rats, aging between 5-6 weeks old, weighing (130 ± 20) g (purchased from Shanghai SLAC Laboratory Animal Co., Ltd., Shanghai, China) were raised in the specific pathogen free (SPF) grade feeding room, which were divided into normal group (*n* = 10), negative control (NC) group (*n* = 10) and liver injury group (*n* = 10). Rats in the liver injury group were fed with 15 mg/kg 2-acetyailamifluorene (2-AAF)/polyethylene glycol (PEG) solution every day. After a week, rats were anesthetized with intraperitoneal injection of 1% sodium pentobarbital followed by 2/3 liver resection and on the second day after operation, rats were continually fed with 15 mg/kg 2-AAF/PEG solution for a week. Rats in the NC group were fed with PEG solution and rats in the NC group and the liver injury group were sacrificed on the 21st day after operation for the collection of liver tissues.

### 4.3. Hematoxylin-Eosin (HE) Staining

Liver tissues were fixed with 4% paraformaldehyde, rinsed with phosphate-buffered saline (PBS), dehydrated in gradient ethanol and immersed in xylene. After the xylene solution is dissolved in distilled water, the tissue was moved into the melted paraffin wax in the incubator which is fully penetrated with paraffin wax. Then, the tissue was removed into the embedding device (containing melted paraffin wax), when the liquid paraffin was cooled into a block, the paraffin block was sliced and dewaxed for preparing tissue sections. The prepared sections were stained with hematoxylin for 5 min, after washing the tissue sections were counterstained with 0.5% eosin for 2 min. Then, gradient alcohol dehydration and xylene transparency were performed and all sections were mounted with neutral balata and observed under the microscope (Olympus Optical Co., Ltd., Tokyo, Japan). 

### 4.4. Cell Culture and Grouping

HPCs line WB-F344 cells were obtained from the Military Medical Science Academy of the PLA. Cells were incubated in Dulbecco’s modified eagle medium (DMEM, Gibico, Grand Island, NY, USA) containing 10% fetal bovine serum (FBS, Gibico, Grand Island, NY, USA) at 37 °C with 5% CO_2_. Lipofectamine 2000 kit (Invitrogen, Carlsbad, CA, USA) was used for transfection of LncRNA-Dreh interference plasmid and blank plasmid (purchased from Promega Corporation, Madison, WI, USA) into WB-F344 cells and cells were divided into si-Dreh group and control group accordingly, the other cells without any transfection were used as the blank group. 

### 4.5. Reverse Transcription Quantitative Polymerase Chain Reaction (RT-qPCR)

Total RNA was extracted from hepatic tissues and liver progenitor cells (after transfection for 48 h) with Trizol reagent. The cDNA template was synthesized by reverse transcription using PCR system (ABI Company, Oyster Bay, NY, USA). RT-qPCR was conducted using ABI 7500 Real-Time PCR system. The primers are shown in Table 1. The β-actin was used as the internal reference and 2^-ΔΔCt^ method was applied for analysis.

### 4.6. Western Blot Analysis

The total proteins were extracted from liver tissues and liver progenitor cells (after transfection for 48 h) and separated in a 10% sodium dodecyl sulfate (SDS)-polyacrylamide gel electrophoresis. After transferring to the polyvinylidene fluoride (PVDF) membrane (Millipore, Billerica, MA, USA), samples were blocked with 0.5% skim milk for 1 h at room temperature. After rinsing, samples were incubated with Vimentin (1:1000), β-actin (1:5000) and the primary antibody rabbit anti-mouse at 4 °C overnight. Then samples were eluted with phosphate buffer saline with Tween20 (PBST) and the secondary antibody goat anti-rabbit (1:10,000) added for 1h incubation at room temperature. All the antibodies above were purchased from Abcam (Cambridge, MA, USA). Then the chemiluminescence reagent was used for developing, the gray value of protein expression was analyzed by Image software. The β-actin was used as the internal reference and the ratio of the density of interest protein was the relative amount of interest protein. 

### 4.7. Cell Counting Kit-8 (CCK-8) Assay

Cells (after transfection for 48 h) from each group were inoculated in a 96-well plate, after the adherent cell growth, the cell number was counted at specific time points. The steps are as follows: After abandoning the culture medium, 100 µL fresh culture solution which contained 10 µL CCK-8 reagent (Beyotime Biotechnology Co., Ltd., Shanghai, China) was added and the plate was placed in a CO_2_ gas incubator for 2 h, the OD of each well was then detected at 450 nm using enzyme-labeling instrument (Bio-Rad, Inc., Hercules, CA, USA). Six repeat wells were set in the experiment.

### 4.8. Transwell Assay

Cells (after transfection for 48 h) were collected to prepare cell suspension with the cell density of 5 × 10^5^ cells/mL and 200 µL was added into the apical chamber of Transwell chamber (Corning Glass Works, Corning, NY, USA) and 600 µL DMEM containing 10% FBS was added in the basolateral chamber. Cultured at 37 °C with 5% CO_2_ for 6 h later, the chamber was removed and cells of the apical chamber were wiped off with cotton bud, then cells were fixed in 4% polyformaldehyde at room temperature for 15 min, washed once with PBS, stained with crystal violet for 10 min and washed again with PBS. The high-power microscope (Olympus Optical Co., Ltd. Tokyo, Japan) were selected and five fields of visions (upper, lower, left, right and middle) were counted to calculate the number of cells. 

### 4.9. Detection of Albumin (ALB) and Alpha Fetoprotein (AFP)

Cells (after transfection for 48 h) from each group were inoculated in culture plates on the 5th, 8th, 11th, 14th and 17th day respectively, the medium was replaced with the basic culture medium (without FBS and other ingredients). Then cells were cultured for 24 h later (on the 6th, 9th, 12th, 15th and 18th day), after centrifugation, the culture supernatant was collected. ALB kit (Rsbio Biological Technology Co., Ltd., Shanghai, China) and AFP kit (Northern Biotechnology Research Institute, Beijing, China) were used to determine the content of ALB and AFP respectively and the culture supernatant was used as the blank control.

### 4.10. Statistical Analysis

The statistical analysis was conducted using SPSS 20.0 (IBM Corp. Armonk, NY, USA). The measurement data were expressed as the mean ± standard deviation. Comparisons between two groups were analyzed using the t-test. One-way analysis of variance (ANOVA) was applied for comparison between multiple groups. Enumeration data were presented in percentage or rate and comparison between groups was performed by chi-square test and *p* < 0.05 was considered statistically significant. 

## 5. Conclusions

In summary, we provide strong evidence that by inhibiting LncRNA-Dreh expression, the Vimentin expression was increased and the ability of proliferation and migration of HPCs was enhanced but unfortunately it can cause the abnormal differentiation of HPCs. Nevertheless, LncRNA-Dreh can be further observed by gene chip technology to detect whether there are any other targeted proteins that can play a role in liver regeneration.

## Figures and Tables

**Figure 1 ijms-20-02549-f001:** Liver tissues are disorderly arranged in rats with liver injury and appeared ductular reaction. HE, hematoxylin-eosin; NC, negative control by polyethylene glycol (PEG) solution. Ten rats in each group. Scale bars represent 25 μm.

**Figure 2 ijms-20-02549-f002:** Liver tissues of rats with liver injury exhibit decreased LncRNA-Dreh. * *p* < 0.05, vs. the normal by water and NC groups by PEG solution. The results are given as means ± SD (*n* = 10 rats). Statistical analyses were performed according to ANOVA with chi-square test (*p* < 0.05).

**Figure 3 ijms-20-02549-f003:** Liver tissues of rats with liver injury exhibited increased Vimentin expression. Panel (**A**), mRNA expression of Vimentin in liver tissue of rats with liver injury measured by RT-qPCR; panel (**B**,**C**), the protein expression of Vimentin in liver tissue of rats with liver injury measured by Western blot; * *p* < 0.05, vs. the normal by water and NC groups by PEG solution. The results are given as means ± SD (*n* = 10 rats). Enumeration data were performed according to ANOVA with chi-square test (*p* < 0.05).

**Figure 4 ijms-20-02549-f004:** The WB-F344 cells proliferation is elevated after silencing of LncRNA-Dreh. * *p* < 0.05, vs. the blank and control groups. Cells without any transfection were used as the blank group, cells with blank plasmid transfection were used as the control group. The results are given as means ± SD (*n* = 3 samples). Enumeration data were performed according to ANOVA with chi-square test (*p* < 0.05).

**Figure 5 ijms-20-02549-f005:** The WB-F344 cells migration is enhanced after silencing of LncRNA-Dreh. * *p* < 0.05, vs. the blank and control groups. (**A**) the Transwell assay of WB-F344 cells proliferation; Scale bars represent 50 μm. (**B**) the number of migrated cells of the blank, control and si-Dreh groups. Cells without any transfection were used as the blank group, cells with blank plasmid transfection were used as the control group. The results are given as means ± SD (*n* = 3 samples). Enumeration data were performed according to ANOVA with chi-square test (*p* < 0.05).

**Figure 6 ijms-20-02549-f006:** The Vimentin expression of WB-F344 cells increases after silencing of LncRNA-Dreh. (**A**) mRNA expression of Vimentin increased in WB-F344 cells after inhibition of LncRNA-Dreh measured by RT-qPCR; (**B**) the density of Vimentin and β-actin in WB-F344 cells of the blank, control and si-Dreh groups; (**C**) relative expression of Vimentin increased in WB-F344 cells after inhibition of LncRNA-Dreh measured by Western blot analysis; * *p* < 0.05, vs. the blank and control groups. Cells without any transfection were used as the blank group, cells with blank plasmid transfection were used as the control group. The results are given as means ± SD (*n* = 3 samples). Enumeration data were performed according to ANOVA with chi-square test (*p* < 0.05).

**Figure 7 ijms-20-02549-f007:** The synthesis ability of ALB and AFP during the differentiation of WB-F344 is changed after LncRNA-Dreh silencing. (**A**) synthesis ability of ALB is decreased during the differentiation of WB-F344 after LncRNA-Dreh silencing; (**B**) synthesis ability of AFP is increased during the differentiation of WB-F344 after LncRNA-Dreh silencing. * *p* < 0.05, vs. the blank and control groups at the 6th, 9th, 12th, 15th and 18th day. Cells without any transfection were used as the blank group, cells with blank plasmid transfection were used as the control group. The results are given as means ± SD (*n* = 3 samples). Enumeration data were performed according to ANOVA with chi-square test (*p* < 0.05).

**Table 1 ijms-20-02549-t001:** Primer sequences for reverse transcription quantitative polymerase chain reaction.

Gene	Primer Sequence
LncRNA-Dreh	5′-GUGCCUGUCACAAACAGAUTT-3′
	5′-AUCUGUUUGUGACAGGCACTT-3′
Vimentin	5′-TGGCACGTCTTGACCTTGAA-3′
	5′-GGTCATCGTGATGCTGAGAA-3′
β-actin	5′-CTACCTCATGAAGATCCTGACC-3′
	5′-CACAGGATTCCATACCCAAG-3′

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
