# Peer review of "Regulation of Long Non-Coding RNA-Dreh Involved in Proliferation and Migration of Hepatic Progenitor Cells during Liver Regeneration in Rats"

_ijms, 2019, doi:10.3390/ijms20102549_

Round 1

Reviewer 1 Report

The authors addressed all of my concerns.

Reviewer 2 Report

The authors replied to the comments and made the minimal changes required. No substantial additional experiments, quantifications and controls have been provided thus making the results unfortunately less convincing that they could be. 

This manuscript is a resubmission of an earlier submission. The following is a list of the peer review reports and author responses from that submission.

Round 1

Reviewer 1 Report

The manuscript described a change of long non-coding RNA-Dreh (LncRNA-Dreh) in a classical liver regeneration model, which is created by 2-acetylaminofluorene (2-AAF) treatment following partial hepatectomy (PHx) in rats. The authors performed both in vivo and in vitro studies to confirm their finding. The study is interesting, and their results could be useful for other researchers. Although it is still a preliminary study without further mechanistically analysis, the manuscript still can be documented in IJMS after a minor revision.

Figure 7. The legend in the figure does not match the description elsewhere.

Reviewer 2 Report

Ruan et al.

IJMS- 483307

Conflict of interest: none

The following report is divided into two parts containing:

A-General comments concerning the global appreciation of the work.

B-Specific comments concerning particular aspects of the article.

A-GENERAL ANALYSIS AND COMMENTS

In this study, the authors investigate the role of the long non-coding RNA-Dreh (LncRNA-Dreh) on the regulation of hepatic progenitor cells in the context of liver regeneration. Data are provided using in vivo model and in vitro experiments, focusing on rats.

Although interesting, there is a lack of experimental strength to support convincing conclusions. Particularly, supporting information regarding experimental validations, controls and statistics including at least 6 experimental measurements should be provided in order to support the conclusions.

B-SPECIFIC ANALYSIS AND COMMENTS

Major comments (5)

Major comments (1/7)

Lines 64-70, please introduce the experiment concept and aims shortly.

Major comments (2/7)

Lines 67-70, the authors describe results regarding structures, size, arrangement, ballooning, number of HPC. Please provide to the reader at least 6 quantification supporting the results section including statistics.

Major comments (3/7)

Figure 1, please briefly provide details on the negative controls in the text section as well as details (time points, lobes) where the sections are done (same comment for figures 2,3, 5 and 6)

Major comments (4/7)

Figure 2, please make the individual dots visible and indicate in the figure legend the number of experiments done as well as the statistical test used (same comment for all figures).

Major comments (5/7)

Figure 3, please correct the mistake in the name of the figure axe 3B “Vimentin". Please make the individual dots visible and indicate in the figure legend the number of experiments done as well as the statistical test used.

Do the authors could provide correlations between the expression of LncRNA-Dreh and Vim supporting an interdependent relationship?

Major comments (6/7)

Paragraph 2.4, please confirm and validate the data by quantifying the proliferation with another method, for example KI67 immunodetection by flow cytometry.

Major comments (7/7)

Figure 4, please provide evidences, controls and validation steps for the experiment in supplementary data (same comment for figure 7). Could the authors confirm theses findings with human material or at least evaluate the specificity of the results with at least another cell type ?

Minor comments (3)

Minor comments (1/3)

Line 37  "Liver regeneration is defined as compensatory hyperplasia" This might be rephrased, depending on the model and trigger; hypertrophia might also be part of the physiological processes. Also more generally, regeneration has been defined as restoration of organ form and function. Other definitions might be given thus suggesting to at least rephrasing as "might be define as"

Minor comments (2/3)

Line 40, "hepatocyte proliferation is damaged" might be rephrase, the proliferation cannot per se be damaged, please clarify.

Minor comments (3/3)

Line 48, please provide references specific for humans.

Reviewer 3 Report

The author preformed silencing experiments of long non-coding RNA-Dreh and evaluated its effects. However, some important points are missing.

1. No detailed explanation of experimental  procedures. What is the difference between "Normal" and "NC"? What kind negative treatment were applied? 

2. How did they silencing lncRNA? KO? RNAi? They also did not confirm how much lncRNA expression was decreased by silencing.

3. There are no direct experimental confirmations of interaction between lncRNA and Vimentin.

The authors add more experiments and explanations of procedures before submitting revised manuscript.